# Feasibility of Mass-Spectrometry to Lower Cost and Blood Volume Requirements for Assessment of B Vitamins in Patients Undergoing Bariatric Surgery

**DOI:** 10.3390/metabo10060240

**Published:** 2020-06-10

**Authors:** Seth Armah, Mario G. Ferruzzi, Nana Gletsu-Miller

**Affiliations:** 1Department of Nutrition, School of Health and Human Sciences, University of North Carolina at Greensboro, 319 College Avenue, 318 Stone Building, Greensboro, NC 27412, USA; s_armah@uncg.edu; 2Department of Food, Bioprocessing and Nutritional Sciences, Plants for Human Health Institute, North Carolina State University, 600 Laureate Way, Kannapolis, NC 28081, USA; mferruz@ncsu.edu; 3Department of Applied Health Science, School of Public Health, Indiana University Bloomington, 1025 E 7th Street, Suite 112B, Bloomington, IN 47405, USA

**Keywords:** B vitamin complex, nutrition assessment, analytical chemistry techniques

## Abstract

Bariatric surgery induces deficiencies in a combination of B vitamins. However, high costs and a large blood volume requirement are barriers to routine screening. We adapted and validated a method coupling tandem mass spectrometry (MS/MS) with high-performance liquid chromatography (HPLC) to facilitate cost-effective analysis for simultaneous detection of B vitamins in low volumes of plasma. Based on existing methods, pooled plasma was extracted using hexane and acetonitrile and seven B vitamin analytes were separated using HPLC. Detection was performed with an Agilent 6460 triple quadrupole tandem mass spectrometer (MS/MS) using electrospray ionization in the positive ion mode. We evaluated linearity, recovery, precision, and limit of detection, as well as costs of the assay. We evaluated seven B vitamins from plasma; five (riboflavin, nicotinamide, pantothenic acid, pyridoxine, and biotin) were detected and quantified with precision and linearity. Recovery ranged from 63 to 81% for each of the vitamins, except for nicotinamide—the recovery of which was suppressed to 40%, due to plasma matrix effects. We demonstrated the feasibility of the HPLC–MS/MS method for use in patients who undergo bariatric surgery by analyzing pooled plasma from patients with a lower cost and blood volume than had we sent the samples to a commercial laboratory. It is advantageous and feasible, in terms of low cost and blood volume requirement, to simultaneously measure plasma concentrations of B vitamins using HPLC–MS/MS. With further improvements, the method may enable personalized nutritional assessment for the nutritionally compromised, bariatric surgery population.

## 1. Introduction

B vitamins include thiamine (vitamin B_1_), riboflavin (vitamin B_2_), niacin (vitamin B_3_), pantothenic acid (vitamin B_5_), pyridoxine (vitamin B_6_), biotin (vitamin B_7_), folate (vitamin B_9_) and cobalamin (vitamin B_12_), which are essential for humans. They are needed in energy and other basic metabolic pathways and red blood cell synthesis, and their deficiency comes with consequences such as anemia and poor nervous system function [1]. Conventionally, B vitamins are quantified using microbiological, immunological or enzymatic assays. These require the separate quantification of each B vitamin, thus requiring a long time for analysis, as well as high costs and large sample volumes. These shortcomings limit nutritional assessment of multiple combinations of B vitamins, which would be needed for populations that may be malnourished in several B vitamins at the same time. Recent efforts to simultaneously measure several B vitamins in a single sample promise to address issues related to limited accessibility. Progress has been made in food and supplement industries in this regard [2]. Nutritional supplements contain high amounts of vitamin analytes, making quantitation fairly easy. In biological specimens, however, simultaneous extraction of multiple vitamins is challenging due to the complexity of sample composition and the relatively smaller concentrations of the vitamins. Chatzimichalakis et al., followed by Giorgi et al., developed a method to separate and simultaneously measure B vitamins in complex biological blood and urine specimens using high-performance liquid chromatography (HPLC) [3,4]. However, the method used photodiode array detectors which are not very sensitive and necessitated having a large sample volume (>1 mL) for correct identification and quantification of the vitamin analytes [4]. An exciting development is to couple HPLC with tandem mass spectrometry (MS/MS) for detection of the vitamin analytes, which is preferable to ultraviolet–visible spectroscopy or photodiode array detectors in terms of sensitivity and specificity [5]. Several research groups have successfully simultaneously measured B vitamins in blood using HPLC–MS/MS [6,7,8,9]. Some groups went further to apply the HPLC–MS/MS methodology to simultaneously quantify B vitamins in patient populations including those with colorectal cancer [10] and neural tube defects [11].

Individuals who have undergone bariatric weight loss surgery are among patients who have a high risk for developing deficiency in a combination of B vitamins, due to reduced dietary intake and intestinal malabsorption after surgery [12]. However, since the prevalence of B vitamin deficiencies is considered to be low because it is less than 15% [13,14,15], to reduce health care costs, expert guidelines do not recommend widespread screening for B vitamin deficiency following bariatric surgery [16]. Consequently, deficiencies in thiamine, pyridoxine, vitamin B_12_, and folate often go undetected, leading to serious neurological complications that are irreversible [17]. The use of HPLC–MS/MS to simultaneously assess B vitamins offers a strategy for screening bariatric surgery patients that is not burdensome because only one blood draw is required. Current methods require a separate blood draw for each vitamin that is analyzed. Moreover, simultaneous assessment using HPLC–MS/MS should be less costly and require less blood than current enzymatic, immunological or chromatographic methods that measure one nutrient at a time [18]. A goal of our research is to improve nutritional outcomes following bariatric surgery and overcoming barriers regarding individualized nutritional assessment by reducing costs and required sample volume would be important steps. Encouraged by our recent experience measuring vitamin D and its metabolites using HPLC–MS/MS [19], the objective of this current study was to adapt an existing HPLC method [3,4] for the simultaneous quantitation of B vitamins in plasma samples by coupling it with MS/MS. The high throughput as well as accuracy and sensitivity of mass spectrometry analysis makes it particularly suitable for nutritional analysis that can be personalized and inform the diagnosis and evaluate the treatment of individuals in terms of B vitamin nutrition. We hypothesized that HPLC–MS/MS would be a low-cost, small sample-volume method, that could be adapted to enable assessment of B vitamins in bariatric surgery patients for researchers or practitioners using precision medicine approaches.

## 2. Results

Demographics of the participants whose plasma samples we pooled and used for the assay were: 75% female, 75% white, age 51.3 ± 4.3 years, and body mass index 38.2 ± 10.3 kg/m^2^ (Table 1). Eighty percent of the sample used a multivitamin/multimineral supplement. We did not find any differences between the participants associated with the pooled plasma sample and a larger biorepository of over 50 participants within our research study (Table 1). As shown in Figure 1, we achieved chromatographic separation from 100 µL of the standard mix and from 100 µL of the pooled plasma sample for the seven vitamin analytes (Figure 1). To assess the validity of our HPLC–MS/MS methodology for simultaneous assessment of various B vitamins, we evaluated linearity, limit of detection, recovery, precision, extraction efficiency, and matrix effects. Linearity assessed by R^2^ found values for riboflavin, biotin, pantothenic acid, nicotinamide, and pyridoxine that were greater than 0.9950, indicating that the regression curves explained nearly all the variability in the B vitamin concentration and the curves are therefore reliable to be used to calculate the B vitamin concentrations (Table 2). However, the linearity of the curves for folic acid and thiamine were poor (R^2^ = 0.9919, and 0.9948, respectively) and thus we did not further validate these two vitamins. In terms of the LOD, the method was able to detect concentrations 100-fold lower than the low reference cutoff for nicotinamide, pantothenic acid and pyridoxine, but not for riboflavin (5-fold lower than the low reference cutoff) and LOD of biotin was 0.4-fold higher than the low reference cutoff (Table 2). To simultaneously analyze five vitamins from 100 µL of plasma required starting with 250 µL of a blood sample, which is 10-fold lower than the ~2500 µL (5 × 500 µL) blood sample needed if the five vitamins were analyzed separately.

The estimates of extraction efficiency, which indicate how much of each vitamin was retained during the extraction step, ranged from 47 to 59% (Table 3). Due to matrix effects, the recoveries of spiked vitamins from plasma samples were enhanced slightly (63 to 81%) for most of the vitamins, except for nicotinamide—the recovery of which was suppressed to 40% (Table 3).

Precision, assessed as interday CV, using the pooled plasma sample, was low (≤20%) for pyridoxine, biotin, nicotinamide, and pantothenic acid, but was very high for riboflavin (60%) (Table 4). Intraday variability was low for all the five B vitamins (<7%).

A summary of the evaluation of the HPLC–MS/MS method is provided in Table 5. Despite issues regarding poor recovery and precision, the HPLC–MS/MS method proved to be sensitive for detecting at least 5-fold below published reference ranges of riboflavin, niacin, pantothenic acid, and pyridoxine. Finally, our HPLC–MS/MS measurement of vitamin B_6_ in the pooled plasma sample (18.7 ng/mL) agreed with that from a commercial assay (16.1 ± 12.4 ng/mL).

The core metabolomics facility at Purdue University charges $35 per sample for using HPLC–MS/MS. To validate the assay, we spent $400 on standards, $1050 to run the calibration curves, and another $7000 to assess recovery, extraction efficiency, precision and the limits of detection, for a total of $8450. To measure 100 samples (which is a typical amount for our research group), we calculated approximately $2000 for a week of personnel costs for a research technician to extract samples and analyze the samples. Thus, the total cost for 100 samples using the HPLC–MS/MS procedure would be $13,950 ($35 × 100 + $8450+ $2000) versus $20,000 for analyzing the samples commercially (5 × $40 per analyte × 100). Thus, it would cost less to analyze the samples than if we had used a commercial laboratory.

## 3. Discussion

In this study, we coupled high-resolution, mass spectrometry with an existing HPLC separation method for assessment of B vitamin concentrations in plasma samples of patients who have undergone bariatric surgery and evaluated the validity, cost and sample volume requirements of the method. In terms of precision, limits of detection, and recovery, the new method enabled sensitive detection of very low concentrations of riboflavin, pantothenic acid, nicotinamide, pyridoxine, and low concentrations of biotin, but not folic acid and thiamine. However, recovery of all but nicotinamide was poor. We demonstrated the existence of plasma-induced matrix effects, which enhanced the detection of pyridoxine, pantothenic acid, riboflavin, and biotin, but suppressed detection of nicotinamide. Our findings illustrate the potential, upon further optimization of the HPLC–MS/MS method, for a research laboratory to simultaneously assess B vitamins from human plasma of bariatric surgery patients, given the reasonable costs and low burden to patients in terms of blood volume required.

The field of B vitamin assessment is moving away from using microbial, immunological and enzymatic assays, because of challenges including low sensitivity, precision and accuracy [20]. Using liquid chromatography for analyte separation overcomes these challenges, with the added advantage of enabling simultaneous determination of multiple B vitamins. However, coupling HPLC with UV fluorescence or an electrochemical device for detection meant that a large specimen volume was required [20]. The advantage of using MS for detection is the high sensitivity and accuracy. However, the high cost and technical skill needed to operate the instrument have been barriers preventing its use except within commercial reference laboratories [20]. Despite the high cost and technical expertise required to use MS instrumentation, the sensitivity, accuracy and high-throughput capability make the methodology ideal for personalized approaches in terms of nutritional assessment. As MS has become more accessible for use in research, we sought to incorporate the technology with the goal of reducing the cost and sample volume required for B vitamin assessment in patients undergoing bariatric surgery. This population is highly susceptible to nutrient deficiencies, but few patients undergo routine screening, because of the high cost. Because of our research on the impact of bariatric surgery on anemia induced by iron and copper deficiencies [21,22], we sought to measure B vitamins as part of a nutritional panel, since deficiencies in many B vitamins lead to this complication. The literature contains recent research on HPLC–MS/MS technologies for simultaneous assessment of B vitamins from human biological fluids including serum/plasma [9,10,11], whole blood [6,7,8], blood spots [23], and breast milk [24]. The analysis in each of these cases required less than 500 µL of blood, which is less than the 1000 µL required for HPLC techniques that did not use mass spectrometry [3]. The standard curves showed linearity range between 5.0 × 10^−3^ to 20,000 ng/mL, a wide range compared to the normal range of B vitamin concentrations found in human plasma, and thus applicable for screening patients who undergo bariatric surgery. For pantothenic acid, for example, the upper limit of the normal range is 147 ng/mL, and the linearity of the assay was valid for concentrations approximately 13.6-fold higher than this. For screening for vitamin deficiencies in the bariatric surgery population, detection of very low concentrations would be important. By comparing normal reference ranges to the LODs reported in our study, we show that very low concentrations of vitamins B_3_, B_5_ and B_6_ in plasma obtained from bariatric patients can be detected using this new method. For vitamin B_6_, for example, the lowest value we have measured in a bariatric surgery patient was 2 ng/mL, well within the assay’s limit of detection of 0.05 ng/mL for that vitamin. On the other hand, more effort is needed to improve the LODs we determined for riboflavin and biotin.

Our research adds to the limited existing literature demonstrating HPLC–MS/MS for simultaneous quantification of B vitamins in small volumes of human plasma, and we also demonstrated the method’s feasibility in terms of the low cost. A challenge we encountered was poor recovery. The main reasons for low recovery could be degradation of the vitamin during storage and the extraction process, which may have occurred for thiamine and folic acid, and ion suppression due to matrix effects, which we observed with nicotinamide. To improve the stability of folate and thiamine, some groups add antioxidants such as ascorbic acid, citric acid and dithiothreitol to the samples during the preparation step [9,10,11]. The choice of biological specimen also impacts recovery due to matrix effects. Consistent with the literature on simultaneous B vitamin assessment when using mass spectrometry [9,10], we observed interference due to the plasma matrix to various extents, leading to enhancement or suppression of the detection of the analyte. The implication of matrix interference is that the same matrix should be used throughout the processes of method validation.

Another matrix issue we may consider when analyzing B vitamins in the future is the suitability of plasma versus whole blood. For example, while Redeuil et al. successfully measured thiamine and its metabolites in plasma [9], other groups preferred using whole blood [6,7,8], since the concentration of thiamine diphosphate in erythrocytes is a better measure of thiamine status. Finally, since we expect to use the HPLC–MS/MS method to comprehensively assess the status of B vitamins in patients undergoing bariatric surgery, we would include assessment of vitamin B_12_ in future analysis.

## 4. Materials and Methods

### 4.1. Standard Preparation

Seven B vitamin standards (thiamine for vitamin B_1,_ riboflavin for vitamin B_2_, nicotinamide for vitamin B_3_, pantothenic acid for vitamin B_5_, pyridoxine hydrochloride for vitamin B_6_, biotin for vitamin B_7_, and folic acid for vitamin B_9_) were purchased from Sigma Aldrich Corporation (St. Louis, MO, USA). To make a 100,000 ng/mL stock B vitamin cocktail, 10 mg of each of the seven different B vitamins were mixed into 100 mL of 1% formic acid/H_2_O. The standard was divided into small tubes for single use, covered with aluminum foil and stored in a freezer at −80 °C. Portions of the stock B vitamin cocktail were later diluted into various concentrations for constructing the standard curves and also for assessing limit of detection.

### 4.2. Sample Extraction

We pooled plasma from a subgroup of four individuals within a biorepository of participants who had undergone bariatric surgery and were being screened for copper and iron deficiency with and without anemia (total population, N = 52) [21]. In the method development and validation for the current study, we used the same pooled plasma sample throughout. All subjects gave their informed consent for inclusion before they participated in the study. The study was conducted in accordance with the Declaration of Helsinki, and the protocol was approved by the IRB at Purdue and Indiana Universities (#1112007617). For this study, we routinely send out specimens to commercial laboratories to measure B vitamins that are also associated with anemia including vitamin B_1_ (measured by Mid America Clinical Laboratories, Indianapolis, IN, USA, using LC–MS/MS, requiring 1 mL serum), vitamin B_6_ (measured using HPLC by Quest Diagnostics, Valencia, CA, USA, requiring 0.5 mL plasma), and folate and vitamin B_12_ (measured using electrochemiluminescent immunoassay by South Bend Medical Foundation, South Bend, IN, USA, requiring 0.6 mL serum). We used the extraction procedure published by Giorgi et al., with minor modifications [4]. To 100 µL of the pooled plasma sample, we added 10 µL of 10 µM ethyl gallate (as internal standard, which also acts as an antioxidant) and 100 µL of N-hexane, and then we vortexed (2 min) and centrifuged (3380× *g*, 2 min). The lower aqueous layer was collected and mixed with 300 µL of acetonitrile and centrifuged (3380× *g*) to obtain the upper layer. We dried the sample using a nitrogen bath and reconstituted the final residue with 100 µL of 1% formic acid.

### 4.3. HPLC–MS/MS

We used chromatographic conditions similar to those published by Giorgi et al. [4], except for substituting trifluoroacetic acid for formic acid. The vitamin separation was achieved using an Agilent Rapid 1200 HPLC system and data collection was carried out using Agilent Mass Hunter software (Agilent Technologies, Santa Clara, CA, USA). The reverse-phase chromatography procedure involved a 2.1 × 3.5 × 100 µm column; temperature, 30 °C; mobile phase A: 18 MΩ H_2_O with 1% formic acid; and mobile phase B: acetonitrile with 0.1% formic acid. We eluted using a combined isocratic and linear gradient at a constant flow rate of 0.31 mL/min as follows: initial conditions, 100% mobile phase A constant for the first 3.5 min; a linear decrease to 50:50 during the next 3.5 min; a linear decrease to 5:95 (A:B) for the next 7 min; and a linear increase to 100% A in the last 5 min. The injection volume was 10 µL, and the needle was washed with 1:1 water: isopropanol, 100% for 10 s between runs to minimize carryover. Detection involved an Agilent 6460 triple quadrupole tandem mass spectrometer using electrospray ionization (ESI) in the positive ion mode (Agilent Technologies, Santa Clara, CA, USA). Details of the mass spectrometry acquisitions are as follows: gas temperature—325 °C, gas flow rate—8 L/min, nebulizer—45 psi, sheath gas heater—250 °C, sheath gas flow—7 L/min, and capillary voltage—4000 V. The collision energies were as follows: riboflavin—20 V, nicotinamide—20 V, pantothenic acid—10 V, pyridoxine—10 V, biotin—15 V, thiamine—10 V, and folic acid—15 V.

### 4.4. Linearity

We assessed linearity by constructing a calibration curve using the B vitamin cocktail solution at eight different concentrations (0, 500, 1000, 2000, 5000, 10,000, 15,000 and 20,000 ng/mL) and reported the coefficient of determination (R^2^) for each curve. Each solution contained the same concentration of internal standard (10 µmolar ethyl gallate), which also acted as an antioxidant.

### 4.5. Recovery

For assessment of recovery, we included estimation of extraction efficiency and matrix effects, as Saar et al. suggested [25], given that blood sample constituents can enhance or suppress ion formation during electrospray ionization. To assess recovery, we spiked the pooled plasma sample with 100 µL of the B vitamin cocktail (5000 ng/mL). The spiked plasma was taken through the entire extraction procedure as described above and the extracted sample was reconstituted with 100 µL of 1% formic acid. In addition to the spiked sample, an unspiked plasma sample from the same pool was also extracted and analyzed for the B vitamin content. The recovery was calculated as the difference between the B vitamin concentration in the spiked and the raw unspiked sample, expressed as a percentage of the concentration of each vitamin obtained from the pure standard solution. The post-extraction spiked sample was prepared by first extracting the pooled plasma sample. After that, 100 µL of 5000 ng/mL B vitamin cocktail standard was added to the remaining residue post-extraction. Extraction efficiency was determined by expressing the concentration of the spiked sample as a percentage of the concentration of the post-extraction spiked sample. Matrix effects were calculated by dividing the vitamin concentration of the post-extraction spiked sample by the concentration of each vitamin obtained from the pure standard solution.

### 4.6. Precision

We repeatedly assessed the same pooled plasma sample to obtain the intraday and interday coefficients of variation (standard deviation/mean). Intraday variability was assessed by analyzing the plasma sample five times, whereas interday variability was assessed by analyzing the sample on three consecutive days.

### 4.7. Limit of Detection

To determine the limit of detection (LOD), as has been described [26], we diluted the stock B vitamin standard solution (100,000 ng/mL) into 17 lower concentrations ranging from 20,000 ng/mL to 5.0 × 10^−3^ ng/mL. For each B vitamin, we then determined the lowest detectable concentration.

### 4.8. Normal Reference Ranges

We compared the values we obtained from the current HPLC–MS/MS method to reference ranges published by established organizations (United States National Academy of Medicine, Washington, DC, USA) and national reference clinical laboratories (Mayo Clinical Medical Laboratory, ARUP Laboratories, Salt Lake City, UT, USA; Quest Diagnostics, Seacaucus, NJ, USA).

### 4.9. Cost Analysis

The cost of B vitamin analysis via HPLC–MS/MS was compared to that for analyzing the blood samples at a commercial laboratory such as Mid America Clinical Laboratories (Indianapolis, IN, USA), Quest Diagnostics (Secaucus, NJ, USA), or ARUP Laboratories (Salt Lake City, UT, USA). We presumed that the user has access to existing HPLC and mass spectrometry instrumentation and did not include these equipment costs. At the mass spectrometry core facility at Purdue University, users pay for each sample they run on the HPLC–MS/MS instrument, and other costs include personnel labor, purchasing the B vitamin standards, as well as the supplies and consumables for plasma extraction. Regarding commercial pricing for B vitamins, we received quotes of $7.00 to $42.50 for analysis of vitamin B_6_, $39.85 for vitamin B_2_, and $73.00 for folate.

### 4.10. Data Analysis

Data are presented as the mean and standard deviation. Standard curves, regression and other data/statistical analysis were performed using Excel software (Microsoft Corporation, Redmond, WA, USA)

## 5. Conclusions

In summary, our current research demonstrated that using HPLC–MS/MS is feasible, in terms of low required cost and sample volume, to simultaneously measure plasma concentrations of five B vitamins in a screen for nutritional deficiency following bariatric surgery. A strength of our research is that we evaluated the method’s linearity, precision, limit of detection, extraction efficiency, and recovery. For the method to be fully applicable for bariatric surgery nutrition, future research needs to improve the recovery of analytes during extraction and detection, and improve the measurement of folate and thiamine, as well as include assessment of vitamin B_12_. Since it is well established that bariatric surgery compromises the status of B vitamins, a low-cost, less-invasive, screening strategy should alleviate barriers to individualized diagnosis and intervention. Therefore, while preliminary, our current research should encourage laboratories to further develop methods for simultaneous assessment of B vitamins which will advance the field of personalized nutrition.

## Figures and Tables

**Figure 1 metabolites-10-00240-f001:**
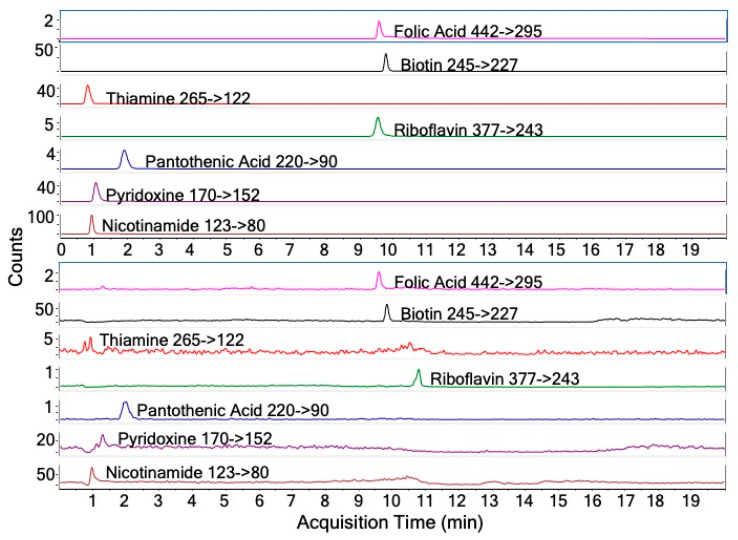
HPLC–MS/MS elution profiles of vitamin analytes. Peaks are labelled with each B vitamin and its specific mass transition (*m*/*z*). The top panel is the pure standard mixture and the bottom panel is a 100 µL pooled plasma sample.

**Table 1 metabolites-10-00240-t001:** Description of participants who provided pooled plasma sample.

Characteristic (N = 4)	Mean (SD)
Age, y	51.3 ± 4.3
Body mass index, kg/m^2^	38.2 ± 10.3
Woman (%)	75
Race White Black	75 25
Use of multivitamin/multimineral supplement (Yes,%)	75
Vitamin B_1_ (ng/mL)	147.0 ± 53.7
Vitamin B_6_ (ng/mL)	16.1 ± 12.4
Vitamin B_9_ (ng/mL)	15.1 ± 7.4

Characteristics of the four participants whose plasma was pooled and analyzed using HPLC–MS/MS; previously obtained measurements of vitamins B_1_, B_6_ and B_9_ from commercial assays are included. Their characteristics were similar to a larger sample of bariatric surgery participants within our anemia research biorepository (N = 52, age = 49.1 ± 8.3, 93% women, body mass index = 32 ± 9.1 kg/m^2^, 80% using multivitamin/multimineral supplements, with commercial assay measurement of vitamin B_6_ = 19.0 ± 22.3 ng/mL [range, 2.0, 163.0 ng/mL]; *p* > 0.05).

**Table 2 metabolites-10-00240-t002:** Validity of the method in terms of standard curve linearity and precision.

Vitamin	Coefficient of Determination (R^2^)	Intraday CV	Interday CV
B_2_ (Riboflavin)	0.9971	6.6	61.5
B_3_ (Nicotinamide)	0.9977	3.7	6.7
B_5_ (Pantothenic acid)	0.9968	3.0	20.7
B_6_ (Pyridoxine)	0.9997	1.0	3.6
B_7_ (Biotin)	0.9963	3.0	4.9

Precision was determined by the intraday and interday coefficients of variation (CV), determined by the standard deviation /mean.

**Table 3 metabolites-10-00240-t003:** Extraction efficiency and percentage recovery estimation for new method.

Vitamin	Raw Sample (ng/µL)	Spiked Sample (Pre-Extraction) (ng/µL)	Spiked Sample (Post-Extraction) (ng/µL)	Pure Vitamin Cocktail (ng/µL)	Extraction Efficiency (%)	Matrix Effects (%)	Recovery (%)
B_2_ (Riboflavin)	0.0199	2.841	5.788	3.724	49.1	155.4	75.8
B_3_ (Nicotinamide)	0.0388	0.971	1.650	2.328	58.8	70.1	40.0
B_5_ (Pantothenic acid)	0.2297	3.301	5.99	3.781	55.0	158.4	81.2
B_6_ (Pyridoxine)	0.0187	2.777	5.306	4.029	52.3	131.6	68.5
B_7_ (Biotin)	0.0549	2.608	5.600	4.086	46.6	137.0	62.5

Descriptions of the raw sample as well as the spiked pre- and post-extraction samples are described in Materials and Methods. Extraction efficiency was calculated by the concentration of the pre-extraction spiked sample as a percentage of the concentration of the post-extraction spiked sample. Matrix effects were calculated by dividing the vitamin concentration of the post-extraction spiked sample by the concentration of each vitamin obtained from the pure standard solution. Recovery was calculated using the difference between the B vitamin concentration in the spiked and the raw unspiked sample, expressed as a percentage of the concentration of each vitamin obtained from the pure standard solution.

**Table 4 metabolites-10-00240-t004:** Assessment of limit of detection and the concentration of pooled plasma sample.

Vitamin	Normal Range (ng/mL) *	Limit of Detection (ng/mL)	Limit of Detection Fold Decrease from Lower Cutoff	Sample Concentration (ng/mL)
B_2_ (Riboflavin)	1–19	0.2	5	19.9
B_3_ (Nicotinamide)	5–75	0.05	104	38.8
B_5_ (Pantothenic acid)	37–147	0.2	185	122.7
B_6_ (Pyridoxine)	5–50	0.05	100	18.7
B_7_ (Biotin)	0.2–3.0	0.5	0.4	7.4

The limit of detection was determined as the lowest detectable concentration from the standard curve. To obtain the fold decrease from the lower limit, we divided the lower cutoff from the reference range by the limit of detection value. The analyte concentrations determined from the pooled sample are provided. * Reference ranges are values published by established organizations (United States National Academy of Medicine) and national reference clinical laboratories (Mayo Clinical Medical Laboratory, ARUP Laboratories, Salt Lake City, UT, USA; Quest Diagnostics, Seacaucus, NJ, USA.

**Table 5 metabolites-10-00240-t005:** Summary of B vitamin analysis of HPLC–MS/MS method.

Vitamin Analyte	High Linearity	High Inter and Intraday Precision	Low Limit of Detection	Matrix Effects	High Recovery
B_1_ (Thiamine)	−	N/A	N/A	N/A	N/A
B_2_ (Riboflavin)	+	+/−	+	enhanced	+
B_3_ (Nicotinamide)	+	+	+	suppressed	−
B_5_ (Pantothenic acid)	+	+/−	+	enhanced	+
B_6_ (Pyridoxine)	+	+	+	enhanced	−
B_7_ (Biotin)	+	+	−	enhanced	−
B_9_ (Folic acid)	−	N/A	N/A	N/A	N/A

We evaluated acceptability in terms of linearity, if the value was >99.5%; precision, if it was <15%; recovery, if it was >75%; limit of detection, if it was 5-fold or more lower than the reference range; and matrix effects as high, if it was >100%; and low, if it was <100%.

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
