# Peer review of "Feasibility of Mass-Spectrometry to Lower Cost and Blood Volume Requirements for Assessment of B Vitamins in Patients Undergoing Bariatric Surgery"

_metabolites, 2020, doi:10.3390/metabo10060240_

Round 1
Reviewer 1 Report
The authors illustrate the use of LC-MS/MS for the simultaneous detection and measurement of plasma B vitamins in bariatric surgery patients. This work presents the possibility to improve the assessment of vitamin B deficiency in terms of analytical throughput and lower sample volumes and poses interesting considerations on the application of this approach for personalized nutritional assessment and intervention in bariatric surgery patients.
This work is well performed and scientifically robust, and can be accepted after minor revisions. Specifically, we suggest to include in the manuscript a discussion of the LOD, LOQ and linearity range of the method in the context of physiologic concentrations of vit B and their changes after/during bariatric surgery treatment. We also suggest to extend the discussion towards the applicability of the method for personalized interventions.
Author Response
Comment 1:The authors illustrate the use of LC-MS/MS for the simultaneous detection and measurement of plasma B vitamins in bariatric surgery patients. This work presents the possibility to improve the assessment of vitamin B deficiency in terms of analytical throughput and lower sample volumes and poses interesting considerations on the application of this approach for personalized nutritional assessment and intervention in bariatric surgery patients.
This work is well performed and scientifically robust, and can be accepted after minor revisions.
We appreciate the reviewer’s constructive comments and by addressing these the manuscript has been improved.
Comment 2: Specifically, we suggest to include in the manuscript a discussion of the LOD, LOQ and linearity range of the method in the context of physiologic concentrations of vit B and their changes after/during bariatric surgery treatment.
Thank you for this recommendation. We used established criteria to evaluate linearity and limit of detection, but we did not highlight how this information relates to applying this method to analysis of bariatric surgery. In the Discussion of revised manuscript, paragraph 2, we have provided text showing that we could measure the ranges of B vitamin concentrations in patients who have bariatric surgery. We state the following:
“The standard curves showed linearity range between 5.0 x 10-3 to 20,000 ng/mL, a wide range compared to the normal range of B vitamin concentrations found in human plasma, and thus applicable for screening patients who undergo bariatric surgery. For pantothenic acid, for example, upper limit of the normal range is 147 ng/mL, and linearity of the assay was valid for concentrations approximately 13.6 times higher than this. For screening for vitamin deficiencies in the bariatric surgery population, detection of very low concentrations would be important. By comparing normal reference ranges to the LODs reported in our study, we show that very low concentrations of vitamins B3, B5 and B6 in plasma obtained from bariatric patients can be detected using this new method. For vitamin B6 for example, the lowest value we have measured in a bariatric surgery patient, was 2 ng/mL, well within the assay’s limit of detection of 0.05 ng/mL for that vitamin. On the other hand, more effort is needed to improve the LODs we determined for riboflavin and biotin.”
Comment 3: We also suggest to extend the discussion towards the applicability of the method for personalized interventions.
We appreciate this recommendation and have added the utility of the new method for personalized medicine. We state the following:
“Since it is well-established that bariatric surgery compromises the status of B vitamins, a low-cost, less-invasive, screening strategy should alleviate barriers to individualized diagnosis and treatment. Therefore, while preliminary, our current research should encourage laboratories to further develop methods for simultaneous assessment of B vitamins which advance the field of personalized nutrition.”
Reviewer 2 Report
The manuscript entitled "Feasibility of Mass-Spectrometry to Lower Cost and Blood Volume Requirements for Assessment of B Vitamins in Patients Undergoing Bariatric Surgery" describes a development and validation of a method for the quantification of some B vitamins in plasma. The reviewer appreciate the large amount of work behind, and thinks it fits the quality and requirements to be considered for publication in this Journal.
I suggest some minor issues to be considered:
I would suggest to use the same units to express concentration through the whole manuscript.
Abstract line 24. Avoid the use of expressions like "good precision and linearity".
2.2. Sample extraction Section. The paragraph is unclear and hard to follow. Please clarify number of samples used, number of individuals , are the patients sent to other labs the same which then are evaluated by your lab? How many samples have been used. Is the pool of plasma the only sample used for the whole method development and validation?
In Table 4. Please check the Normal range for Biotin.
Figure 1. Thiamine, Pyroxamine and Nicotinamide in the lower part of the graph show very low signal to noise ratio, hard to see because is very small, but it's probably very close to limit of detection (S/n >3), and far from limit of quantification (S/N >10). Surprisingly, in the text and tables the range of these compounds is far higher from the limit of detection.
Have the authors tried to use any antioxidants for the sample preparation?
How does this method lower the invasiveness in patients as stated in the conclusions?
Author Response
Comment 1: The manuscript entitled "Feasibility of Mass-Spectrometry to Lower Cost and Blood Volume Requirements for Assessment of B Vitamins in Patients Undergoing Bariatric Surgery" describes a development and validation of a method for the quantification of some B vitamins in plasma. The reviewer appreciate the large amount of work behind, and thinks it fits the quality and requirements to be considered for publication in this Journal.
Response 1: We thank this reviewer the positive comments as well as suggestions for improving it further.
Comment 2: I suggest some minor issues to be considered:
I would suggest to use the same units to express concentration through the whole manuscript.
Response 2: Yes, we have made changes throughout the manuscript and use the unit of ng/mL for all the concentrations measures.
Comment 3: Abstract line 24. Avoid the use of expressions like "good precision and linearity".
Response 3: We agree with this recommendation, and removed the word “good”. We no longer have the phrase “good precision and linearity”.
Comment 4: 2.2. Sample extraction Section. The paragraph is unclear and hard to follow. Please clarify number of samples used, number of individuals , are the patients sent to other labs the same which then are evaluated by your lab? How many samples have been used. Is the pool of plasma the only sample used for the whole method development and validation?
Response 4: We strive to clarify any text that is unclear. We have addressed each of the suggestions in the Method sections in the text below:
“We pooled plasma from a subgroup of four individuals within a biorespository of participants who had undergone bariatric surgery, and were being screened for copper and iron deficiency with and without anemia (total population, N = 52) [20]. In the method development and validation for the current study we used the same pooled plasma sample throughout.” Additional information on the study participants in included the legend of Table 1: “Their characteristics were similar to a larger sample of bariatric surgery participants within our anemia research biorepository (N = 52, age = 49.1 ± 8.26, 93% women, body mass index = 32 ± 9.1 kg/m2, 80% using multivitamin/multimineral supplements, with commercial assay measurements of vitamin B6 = 19.0 ± 22.3 ng/mL [range, 2.0, 163.0 ng/mL]; p > 0.05).”
Comment 5: In Table 4. Please check the Normal range for Biotin.
Response 5: Thank you for finding this error in Table 4, we double-checked that reference range published from Arup Laboratories. While the values are correct in the manuscript, we corrected the range so that the lower cut-off is listed first.
Comment 6: Figure 1. Thiamine, Pyroxamine and Nicotinamide in the lower part of the graph show very low signal to noise ratio, hard to see because is very small, but it's probably very close to limit of detection (S/n >3), and far from limit of quantification (S/N >10). Surprisingly, in the text and tables the range of these compounds is far higher from the limit of detection.
Response 6: We have tried to address the good comment about the bottom panel of Figure 1. As noted by the reviewer, for the plasma sample, the signal to noise ratio is low for thiamine and other evaluation parameters of thiamine were unsatisfactory so we did not evaluate it further. For pyridoxine and nicotinamide the S/N ratio is higher (compared to the profile for thiamine) and we did evaluate better parameters in the measurement of these nutrients. We believe that the image represents the data and we hope the reviewer agrees.
Comment 7: Have the authors tried to use any antioxidants for the sample preparation?
Response 7: The reviewer raises a good point that use of antioxidants would protect the stability of the vitamin analytes. We mentioned in the Discussion section, paragraph 3, that some groups have used ascorbic acid, citric acid or dithiothreitol in the sample preparation. We did not use these antioxidants, but did use ethyl gallate, which has antioxidant activity. Based on the reviewer’s suggestion, we added this information to text in the Methods section:
“To 100 µL of the pooled plasma sample, we added 10 µL of 10 µM ethyl gallate (as internal standard which also acts as an antioxidant)”.
Comment 8: How does this method lower the invasiveness in patients as stated in the conclusions?
Response 8: We mean to say that simultaneous assessment of B vitamins is less invasive or burdensome to patients since less blood is required; current methods require more blood often with a separate blood draws for each vitamin that is analyzed. To clarify this issue, we added the following to the Introduction section, paragraph 2:
“The use of HPLC-MS/MS to simultaneously assess B vitamins offers a strategy for screening bariatric surgery patients that is not burdensome because only one blood draw is required. Current methods require a separate blood draw for each vitamin that is analyzed..”